# PDC Glass/Ceramic Coatings Applied to Differently Pretreated AISI441 Stainless Steel Substrates

**DOI:** 10.3390/ma13030629

**Published:** 2020-01-31

**Authors:** Milan Parchovianský, Ivana Parchovianská, Peter Švančárek, Günter Motz, Dušan Galusek

**Affiliations:** 1Centre for Functional and Surface Functionalised Glass, Alexander Dubček University of Trenčín, Študentská 2, 911 50 Trenčín, Slovakia; ivana.parchovianska@tnuni.sk (I.P.); dusan.galusek@tnuni.sk (D.G.); 2Joint Glass Centre of the Institute of Inorganic Chemistry Slovak Academy of Sciences, Alexander Dubček University of Trenčín and Faculty of Chemical and Food Technology Slovak University of Technology, Študentská 2, 911 50 Trenčín, Slovakia; peter.svancarek@tnuni.sk; 3Ceramic Materials Engineering (CME), University of Bayreuth, D-95440 Bayreuth, Germany; guenter.motz@uni-bayreuth.de

**Keywords:** cleaning, bond coat, PDC coatings, fillers

## Abstract

In this work, the influence of different cleaning procedures on adhesion of composite coatings containing passive ceramic and commercial glasses was investigated. Two compositions (C2c, D2-PP) of double-layer polymer-derived ceramic (PDC) coating systems, composed from bond coat and a top coat, were developed. In order to obtain adherent coatings, stainless steel substrates were cleaned by four different cleaning procedures. The coatings were then deposited onto the steel substrate via spray coating. Pretreatment by subsequent ultrasonic cleaning in acetone, ethanol and deionised water (procedure U) was found to be the most effective, and the resultant C2c and D2-PP coatings, pyrolysed at 850 °C, indicated strong adhesion without delamination or cracks, propagating at the interface steel/bond coat. In the substrate treated by sandblasting and chemical etching, small cracks in the bond coat were observed under the same pyrolysis conditions. After oxidation tests, all coatings, except for those subjected to the U-treated substrates, showed significant cracking in the bond coat. The D2-PP coatings were denser than C2c, indicating better protection of the substrate.

## 1. Introduction

Because of the increasing costs for metals, there is an effort made to enhance the service life of steel components exposed to aggressive environment, which is commonly used in exhaust gas elements, waste incineration plants or in chemical industry. Refractory stainless steels are highly oxidation and corrosion resistant materials. As metal wear and oxidation/corrosion cause significant economic losses, the development of thermal (TB) and environmental barrier coatings (EBC) is the matter of significant importance.

Due to their extraordinary properties at high temperatures and in chemically aggressive environments, non-oxide and oxide PDC ceramic coatings are suitable for increasing the oxidation and corrosion resistance of metals [1]. Preceramic polymers offer a lot of processing advantages that are not possible with traditional ceramics [2]. For example, organosilicon polymer precursors such as polysiloxanes [3], polycarbosilanes [4], or polysilazanes [5,6,7], represent a class of hybrid materials which, by suitable heat treatment (pyrolysis in a controlled atmosphere), provide high purity ceramic materials with an adaptable chemical composition and a well-defined structure. These polymers are characterised by an inorganic polymer chain composed of silicon atoms and organic substituents attached to the backbone. Polysilazanes are currently used as precursors for the synthesis of Si_3_N_4_ and SiCN ceramics, mainly due to the high ceramic yield after pyrolysis (often >80 wt. %) [8]. Polysilazanes are suitable materials for the preparation of protective coatings due to their excellent oxidation and corrosion resistance, UV stability and high hardness. These polymers have excellent adhesion to a wide variety of different substrates, e.g., metal, composite, graphite and glass. The PDC route provide the application of liquid or diluted polymers by easily scalable methods, such as dip-coating [9,10], spin-coating [11], doctor-blade method [12] or spray-coating [13,14]. The choice of a particular method depends on the future use of the coating, the type and shape of the body to be coated and the deposited layer, the size of the covered area, the thickness of the coating and its desired properties.

The main disadvantage of the organosilicon polymer precursors, however, is shrinkage, often more than 50 vol. %, that occurs during the transformation from polymer to an amorphous ceramic [15]. The undesirable shrinkage of the polymer leads to crack formation and, in extreme cases, complete failure of the coating. To overcome these unwanted problems, the coatings that consist of only liquid polymer have to be loaded with beneficial components called fillers. The fillers are active [16,17] or passive, and include a large variety of materials, including YSZ [18], Si_3_N_4_ [19], Al_2_O_3_ [20] and NbC [21] or commercial glasses [22]. The fillers partially or completely compensate the shrinkage, close the pores and increase the coating thickness [23].

The main function of passive fillers is to decrease the bulk fraction of the polymer used, to reduce the amount of gases generated during pyrolysis and, consequently, to alleviate the overall weight loss and shrinkage, and to eliminate the presence of macro-defects by filling the void space in the material without changing its volume. The glass fillers account for densification and sealing of the system, increasing the efficiency of EBC [24]. The service temperature and softening point of the glass filler particles should be matched to increase efficiency of the coating and, in optimum case, heal any defects formed during the coating operation. In our previous work [25], composite PDC coatings with passive fillers and commercial glasses have been developed. Despite using a range of passive fillers, the bulk shrinkage of the polymer precursor has in many cases led to the preparation of porous coatings. Also, the coating often delaminated from the metal substrate, and lost its protective action as EBC.

Another factor ensuring good adhesion of PDC coatings is based on providing an appropriate surface of stainless steel substrate. A number of various pretreatment procedures, such as sandblasting or etching the substrate by different chemical agents, have been described.

In this work, the influence of pretreatment of the AISI441 steel substrate such as sandblasting, etching of the surface or combination of these methods, were investigated in order to choose the most effective type of cleaning and prevent the delamination of the bond coat from the steel substrate. The oxidation tests were performed in order to evaluate the adhesion of the bond coat at higher temperature and longer operating times.

## 2. Materials and Methods

The preparation of the PDC coating systems consisted of 3 steps: (1) synthesis of passive fillers with compositions in the Al_2_O_3_-Y_2_O_3_-ZrO_2_ (AYZ) system by sol–gel Pechini method [26], (2) pretreatment of stainless steel by different methods and (3) preparation of double layer coatings consisting of the bond coat and top coat using a combination of a commercial polymer with passive and glass fillers.

### 2.1. Preparation of the Precursor Powder

A powder in the AYZ system with the composition (in mol. %) 61.49 Al_2_O_3_, 18.51 Y_2_O_3_ and 20 ZrO_2_ was used as a passive filler was prepared by the modified sol–gel Pechini method [27]. Y_2_O_3_ (99.9%, Treibacher Industrie AG, Althofen, Austria) was converted into nitrate by dissolving powder oxide in concentrated HNO_3_ (65% Centralchem, Bratislava, Slovakia). Al(NO_3_)_3_·9H_2_O (p.a., Centralchem, Bratislava, Slovakia) and ZrOCl_2_·8H_2_O (99.9%, Sigma-Aldrich Co. LLC., Darmstadt, Germany) dissolved in deionised water were then added to yttrium nitrate solution. A 1:1 molar ratio solution of C_6_H_8_O_7_ (99.8%, Centralchem, Bratislava, Slovakia) and C_2_H_4_(OH)_2_ (99%, Centralchem, Bratislava, Slovakia) in deionised water was then added dropwise to the mixture, which was then refluxed under a condenser and heated in an oil bath at a temperature of 85–95 °C for 2 h. Subsequently, the solvent was evaporated under continuous stirring. The product was dried, calcined at 850 °C to a white powder and sieved through a 40 µm sieve. For better usability in relatively thin coatings, the AYZ powder was homogenised and granulated by a freeze-drying process. A flowchart of the process of preparation of AYZ powder is presented in Figure 1.

### 2.2. Pretreatment of the Stainless Steel

Ferritic refractory stainless steel grade AISI441, which is commonly used in exhaust gas elements, was used as the metal substrate. Prior to cleaning, the steel sheets were cut into 1 × 1.5 cm^2^ plates to make the samples suitable for further characterisation and testing and to prevent damage to the coated samples by further cutting. This was followed by grinding and chamfering the edges and corners of each sample with sandpaper. To produce adhesive coatings without failure, the surface of stainless steel was treated and cleaned to achieve adhesive coatings with sufficient protective capability at temperatures up to 1000 °C. Four different cleaning procedures were applied, i.e., subsequent ultrasonic cleaning in acetone, ethanol and deionised water; sandblasting with glass beads; chemical etching with Kroll’s reagent; and a combination of the last two methods. The description of cleaning procedures of the steel is summarised in Table 1.

### 2.3. Preparation of the Coatings

A two-layer PDC coating, composed of a bond coat and a ceramic top coat, was applied. The bond coat was prepared from the commercial polymer Durazane 2250 (Merck KGaA, Darmstadt, Germany) by the dip-coating method (dip-coater RDC 15, Relamatic, Glattburg, Switzerland). The pyrolysis of the bond-coat was carried out in air (Nabertherm^®^ N41/H, Nabertherm, Lilienthal, Germany) at a temperature of 450 °C for 1 hour, with heating and cooling rates of 3 K/min. The top coats were prepared from the commercial polymer—Durazane 1800 (Merck KGaA, Darmstadt, Germany), passive fillers and commercial glass. ZrO_2_ stabilised with 8 mol. % Y_2_O_3_ (8YSZ, Inframat^®^ Advanced Materials^TM^, Manchester, CT, USA), AYZ powder prepared by Pechini method and a commercial glass (G018-281, Schott AG, Mainz, Germany) were used as passive fillers. The basic properties of the filler materials are listed in the Table 2.

Commercially available glass was selected to form a viscous melt at the application temperature of the layers, thereby ensuring the healing of any defects and strengthening of the ceramic top layer. The combination of a liquid commercial polymer Durazane 1800 with glass frits and passive filler particles offers the possibility of designing a large range of compositions. Therefore, the composition of the top layer was designed to match the coefficient of thermal expansion (CTE) of the steel substrate and to reduce the CTE mismatch and increase the compatibility of the metal with the ceramic coating. The CTE of stainless steel was provided by the manufacturer (11.5 × 10^−6^/K). The CTE of the prepared coatings were estimated by the rule of mixtures using the CTE of Durazane 1800 (3.0 × 10^−6^/K), 8YSZ (11.5 × 10^−6^/K), AYZ (8.6 × 10^−6^/K) and glass G018-281 (12.1 × 10^−6^/K). Two compositions of top coat were tested, in the following text denoted as C2c and D2-PP. The prepared compositions are listed in Table 3.

In the case of the composition C2c, ZrO_2_ stabilised with 8 mol. % Y_2_O_3_ and glass frits were homogenised in a solution of di-n-butyl ether (Acros Organics BVBA, Geel, Belgium) and dispersant (DISPERBYK 2070, BYK-Chemie GmbH, Wesel, Germany). To break up the agglomerates, the suspensions were dispersed in the ultrasound and homogenised for 48 h by stirring with a magnetic stirrer. Subsequently, Durazane 1800 polymer, with 3 wt. % of dicumyl peroxide (DCP) (Sigma-Aldrich Co. LLC., Darmstadt, Germany), was added to the slurry, which was homogenised for an additional 48 hours in a plastic jar with ZrO_2_ balls (Ø1 mm). After homogenisation, the suspension was applied to the stainless steel with a bond coat by a spray-coating technique from both sides. The suspension was deposited onto steel substrates by spray coating using a spray coater model 780S-SS (Nordson EFD, East Providence, RI, USA). The nozzle diameter of spray gun was 0.71 mm (0.028”). The final suspension was sprayed under the air pressure of 2.2 bar. The distance between the spray gun and the sample was 10 cm. In the composition D2-PP, the AYZ powder prepared by the modified Pechini sol–gel method was used as an additional passive filler. The coated samples were then pyrolysed in air at 850 °C for 1 hour, at a heating and cooling rate of 3 °C/min. A flowchart of the coating processing is presented in Figure 2.

### 2.4. Characterisation Methods

X-ray powder diffraction analysis was used to assign the phase composition of the prepared AYZ powder. Diffraction records were measured on an Empyrean DY1098 powder diffractometer (Panalytical, B.V., Almelo, The Netherlands) with a Cu anode and with X-ray wavelength of λ = 1.5405 Å over 2θ angles of 10–80°. Diffraction records were then evaluated using HighScore Plus (v. 3.0.4) using COD2019 (Crystallographic Open Database). Mean Roughness Depth (Rz) was measured using atomic force microscopy (AFM, Brooker Innova, Billerica, MA, USA). Rz was calculated by measuring the vertical distance from the highest peak to the lowest valley within five sampling lengths, then averaging these distances. The surface morphology of pretreated samples was examined by scanning electron microscopy (SEM, JEOL JSM 7600 F, JEOL, Tokyo, Japan). For detailed examination of the coating/metal interface, the cross sections were prepared via mounting in resin followed by grinding and polishing. The inspection of the coatings was then performed using an SEM equipped with an energy-dispersive X-ray spectroscopy (EDXS) detector (Oxford instruments, Abingdon, UK) and was focused on the evaluation of adhesion, homogeneity and possible failures of the coatings.

### 2.5. Oxidation Tests

The oxidation tests were carried out in a high temperature horizontal electric tube furnace (Clasic 0213T, Clasic, Praha, Czech Republic) in flowing atmosphere of synthetic air (purity 99.5, SIAD Slovakia spol. s.r.o., Bratislava, Slovakia) at a temperature of 950 °C with a heating rate of 5 °C/min and an exposure time of 48 h. The composition of synthetic air is as follows; nitrogen (78%), oxygen (21%), argon (0.9%) and other gases (0.1%).

## 3. Results and Discussion

### 3.1. Characterisation of the AYZ Filler

To achieve a homogenous precursor powder to be used as a filler in the prepared coating, the AYZ powder was prepared via a modified Pechini method [27]. The main advantage of this method is that the metallic ions are immobilised in a rigid polymer network, which ensures their homogeneous dispersion on the atomic scale without precipitation or phase segregation. This process allows complete control over the product stoichiometry, even for more complex oxide powders [27]. To facilitate the use of the AYZ powder in the coatings, the powder was refined by milling and freeze drying to avoid agglomeration of precursor powder and achieve the particle size below 10 μm. SEM micrographs of the AYZ powder after freeze drying are shown in Figure 3. From the as-prepared AYZ powder consists of irregular and angular particles resulting from the crushing process, with the sizes ranging from a few to several tens of micrometres.

The XRD pattern of AYZ powder after calcination is shown in Figure 4, confirming the presence of t-ZrO_2_, as well as yttrium aluminium garnet (Y_3_Al_5_O_12_, YAG) as the main crystalline phases formed during calcination. Smaller amount of the mellilite Y_2_Al_2_O_6_ phase was also observed after calcination. High background in the diffraction pattern indicates the crystalline phases were embedded in an amorphous (glassy) matrix.

### 3.2. Treated Steel Surfaces

The metal surface quality is significant characteristic influencing the adhesion of the protective coating. In the case of double-layer PDC coatings, the weakest point is usually the interface between the bond coat and the metal substrate, due to the presence of impurities or defects on the steel surface, which often lead to delamination of the bond coat during pyrolysis or after corrosion tests. To ensure a high adhesion of the bond coat with the steel, the stainless steel substrate has to be cleaned properly to degrease its surface and remove possible contaminants. The influence of different cleaning procedures on the surface morphology was therefore examined. The Mean Roughness Depth (Rz) was determined using AFM and the surface morphology of pretreated stainless steel samples was examined by SEM. Figure 5 presents the Rz of variously pretreated stainless steel surfaces. The AFM analysis confirmed that different treatment created different sizes of roughness: there is a relation between surface topography and the type of the used cleaning procedure. The Rz was in the range of 0.26 to 1.69 µm. A roughness similar to that observed for untreated surfaces was measured after ultrasonic cleaning, while sandblasting increased the roughness substantially. An increase of Rz was expected to result in stronger bonding at the steel-bond coat interface.

AFM images of the stainless steel cleaned by different methods are shown in Figure 6. SEM-micrographs featuring surfaces of the stainless steel cleaned by different methods are shown in Figure 7. Chemical etching resulted in a surface with irregular topography, with slight roughening of the surface compared to ultrasonically cleaned substrates. More uniform and regular surfaces were obtained by sandblasting with glass beads, or a combination of sandblasting and chemical etching with Kroll’s reagent. These treatments resulted in rough surfaces with rounded edges, with the average surface roughness 1.69 µm for sandblasted samples. Chemical etching of sandblasted substrates slightly decreased the average surface roughness, but the sandblasting induced roughening was still significant.

### 3.3. Characterisation of the Coatings

The SEM cross-sectional micrographs of compositions C2c and D2-PP were obtained through the metal–ceramic interface to investigate the bonding between the bond coat and variously treated steel substrates (Figure 8 (C2c composition) and Figure 9 (D2-PP composition)). The weakest location in a typical double layer coating is usually the interface between the stainless steel and the bond coat, where the cracks or spallation can occur. The spallation of the bond coat is usually caused by thermal and elastic mismatch between the steel and bond coat, due to the presence of impurities at the steel surface, by changing the chemistry of the steel by chemical etching, or growth of stresses followed by the formation of thermally-grown oxides, mostly due to the weak adhesion of bond coat to steel. Irrespective of the applied surface treatment the bond coat did not delaminate from the steel surface after pyrolysis of the coatings at 850 °C, indicating its good adhesion. An undamaged bond coat approximately 1 µm thick was observed in all cases, which acts as an effective diffusion barrier to oxidation during pyrolysis. Only for the D2-PP coating deposited at the substrate etched by Kroll’s reagent the bond coat peeled off from the surface. If the stainless steel was treated by sandblasting or chemical etching or their combination, a few small cracks (marked with white circles) were generated in the bond coat, perpendicular to the substrate surface. The crack formation was attributed to strong adhesion of the bond coat to the metal substrate and, at the same time, by the high volume shrinkage of the polysilazane during heat treatment. No crack formation in the bond coat was observed on samples pretreated via ultrasonic cleaning in acetone, ethanol and deionised water. Cracking could also occur due to uneven and rough sandblasted surface, with sharp edges and peaks acting as stress concentrators in the coatings [24]. The pretreatment by ultrasonic cleaning in acetone, ethanol and deionised water was found to be the most effective process, since no spallation or cracking was observed in the cross section of the bond coat after pyrolysis. Moreover, the ultrasonically cleaned steel surface was uniformly covered by the approximately 1 µm thick bond coat.

In addition, EDXS mapping was carried out on the cross section of D2-PP-coated steel cleaned by ultrasonic treatment in acetone, ethanol and deionised water to demonstrate the existence of an interface bond metal/base coat and base/top coat. EDXS element maps are shown in Figure 10. The bond coat contains mainly Si and O, since during pyrolysis in air the Durazane 2250 was converted to SiO_2_, as confirmed by SEM/EDXS measurement. The presented element maps of Si confirm the formation of the protective bond coat at the steel surface in agreement with the literature [15]. The bond coat enhances the bonding between the steel and the top coat, and it preserves the steel from oxidation/corrosion [15,27]. On steel exposed to ambient environment, a natural oxide layer with adsorbed water is always present. Because of the reactivity of Durazane 2250 with surface-bound –OH groups, steel forms direct metal–O–Si chemical bonds with the base coat, leading to excellent adhesion [28]. The reaction of Durazane 2250 with hydroxyl groups of the substrate surface is described by the following simplified reaction equations [29]:Fe-OH + ≡Si-NH-Si≡ → ≡Fe-O-Si≡ + H_2_N-Si≡(1)
Fe-OH + H_2_N-Si≡ → ≡Fe-O-Si≡ + NH_3_(2) and a direct chemical metal–O–Si bonds between the steel and the precursor-derived coatings are formed. Therefore, the adhesive strength of the PDC coatings to the metal surface is very strong. In the case of sandblasting, we assume that it was the surface roughness of the steel that caused the coating delamination, as sandblasting should not affect the concentration of hydroxyl groups present at the steel surface. However, strong adhesion due to formation of the metal–O–Si bonds causes immobility of the coating during pyrolysis, which does not allow the coating to adjust to volume shrinkage of the steel substrate. Moreover, the sharp borders and peaks of the substrate initiate the formation of cracks perpendicular to the substrate surface. The corrosion/oxidation medium is thus able to penetrate through the cracks to the metal surface causing coating delamination. No crack formation in the bond coat was observed in samples pretreated via ultrasonic cleaning. In the case of chemical etching, we assume that this treatment negatively influenced the adhesion of the bond coat because the concentration of hydroxyl groups on steel surface was significantly affected by etching.

The main elements of stainless steel Fe and Cr were detected below the protective bond coat. The main component of the bond was Si. The top coat consisted of Zr, Al, Y (from AYZ powder) and Ba (from commercial glass).

The top coat layers of both studied compositions—C2c (Figure 8) and D2-PP (Figure 9)—were almost dense, containing only small closed pores present predominantly at the boundaries between the filler particles and the matrix. The filler particles were well coated with the Durazane 1800 precursor, which builds up the matrix, and acts also as an adhesive between the particles. In the case of the C2c coating, all pores were almost spherical. This indicates that the closed pores could result from bubble formation due to the release of dissolved gases, as well as the expansion of insoluble gases (e.g., oxygen or air) entrapped in the initial pores. The pores could result also from the release of gases such as NH_3_, CH_4_, and H_2_ generated during the polymer to ceramic transformation [15]. Note that pore formation cannot be completely avoided when passive fillers are used in the processing of PDC coatings, and some residual porosity usually remains in the final ceramic top coats. Existing pores provide a transportation path for gaseous products of decomposition escaping the coating. In the case of the D2-PP coating, the addition of the AYZ powder with irregular and angular particles have helped to create a rigid and articulate structure. This structure allowed outgassing of the preceramic polymer pyrolysis products from the system thereby effectively reducing the size and amount of pores. The elimination of larger pores, and thus an increased density, led to a significantly more compact coating in comparison to the C2c composition pyrolysed under the same conditions. Although the top layers were not completely dense, microstructure with residual porosity was beneficial for the thermal stability of coatings, and contributed to the mitigation of residual stresses during the heating and cooling cycles [30].

The mismatch of the CTE between the steel and the coating, together with the Young modulus, are critical factors for the resistance of coatings exposed to high temperatures [31]. By reducing the non-conformance of the CTE and the elastic modulus, the tension in the coating is reduced, resulting in a higher stability of the coatings during thermal loading. Moreover, glasses are suitable material for sealing in PDC coatings. By decreasing both quantities (CTE mismatch and Young modulus), the stresses in the layers can be reduced, leading to better coating thermal stability during the oxidation tests. Coatings with both ceramic and glass fillers should then exhibit better thermal compatibility with the steel substrate than the coatings with active and passive fillers without glass tested in our previous work [26]. As a result, no cracks were observed in top coats, preventing penetration of oxidation agents through the coating and attack of the steel substrate.

Static oxidation tests were performed to assess the efficiency of the selected types of steel substrate cleaning. The aim of the oxidation tests was to determine whether the tested cleaning methods would ensure sufficient adhesion of the bond coat to the metal substrate not only during pyrolysis, but also at long-term operating at high temperature. The tests were performed in the atmosphere of synthetic air at 950 °C at a heating rate of 5 °C/min, with maximum duration of 48 hours. SEM cross-sections of C2c and D2-PP compositions after oxidation tests are shown in Figure 11 and Figure 12, respectively. In all cases, except for ultrasonic cleaning, spallation of the bond coat from the surface of stainless steel was observed. Chemical etching of steel with the Kroll’s solution probably caused changes in the microstructure and chemical composition of the steel, resulting in peeling of the coating. For sandblasted samples, or a combination of sandblasting and etching, detachment of the bond coat resulted from high surface roughness: as discussed above, strong adhesion of the coating on a rough surface with sharp edges resulted in formation of cracks perpendicular to the substrate. The cracks serve as a gateway for inward penetration of oxygen, which directly attacks the substrate, creating a layer of oxidation products, and eventually leading the detachment of the bond coat from the substrate. Simple cleaning by ultrasonication in acetone, ethanol and deionised water was found to be the most effective way for achieving a sufficient bonding of the bond coat to the steel substrate after oxidation tests.

Originally, numerous pores were observed across the whole cross-section of the as-prepared PDC glass ceramic coatings. After oxidation tests, a significant increase in the porosity of the layers accompanied by the growth of pores and a decrease of the coating thickness was observed in C2c coating. Decrease of the layer thickness was attributed to differential sintering of individual components (mainly glass fillers) by viscous flow, complemented by inadequate removal of gases entrapped in the ceramic matrix. Due to the softening of the used glass fillers, the cracks in the layers gradually heal, indicating at least partial protection of the metal substrate. After oxidation tests, the D2-PP composition exhibited a compact structure with no increase of the size and amount pores. No cracking or delamination from bond coat was observed.

## 4. Conclusions

Suitable pretreatment of steel substrate, as well as the using a Durazane 2250 bond coat, are prerequisites for preparation of adherent composite coatings. The most effective cleaning process is a 3-step ultrasonic cleaning in acetone, ethanol and deionised water. Small cracks in the bond coat perpendicular to substrate were observed after pyrolysis in bond coats deposited at substrates treated by sandblasting and chemical etching. After oxidation tests, all coatings, except for those applied to substrates cleaned in an ultrasonic bath, delaminated or showed significant cracking of the bond coat. Combination of PDC with tailored fillers and glass systems enable the processing of dense and crack free coating system on stainless steel. Irregular and angular filler particles favour outgassing of the coating during pyrolysis, reducing the total porosity in the layer, and conferring better protection of the substrate against oxidative environment.

## Figures and Tables

**Figure 1 materials-13-00629-f001:**
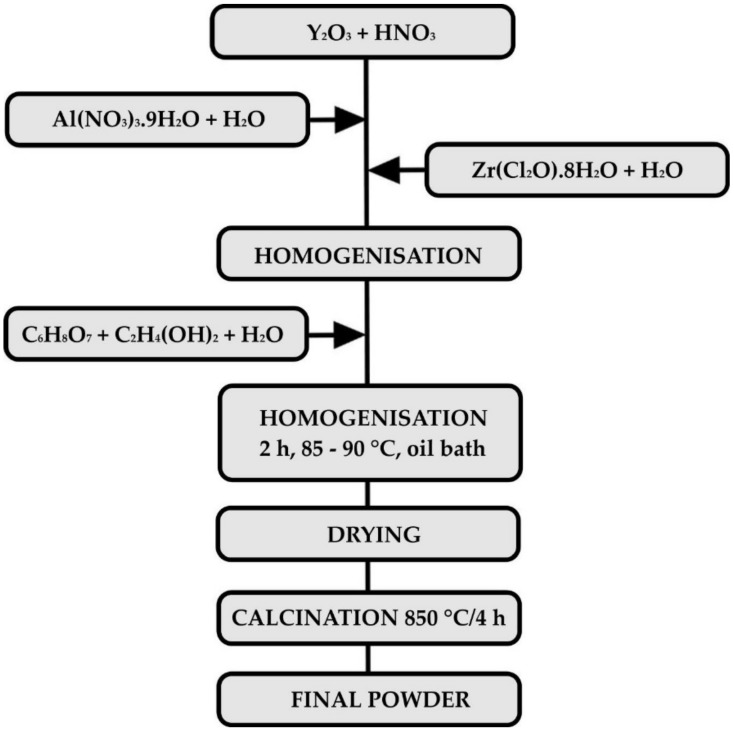
The flowchart of the processing of AYZ powder.

**Figure 2 materials-13-00629-f002:**
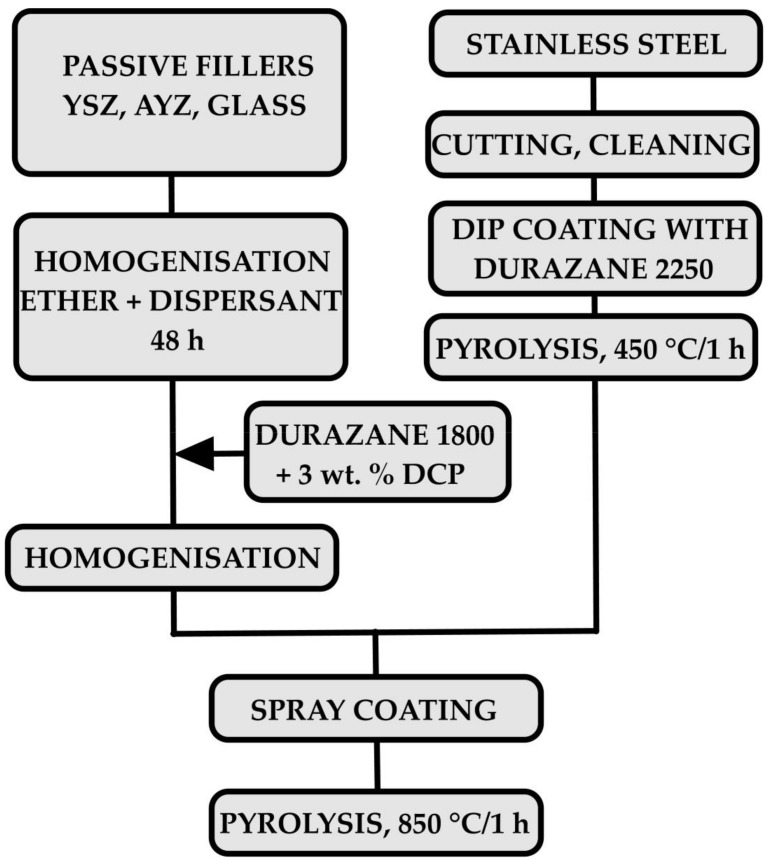
The flowchart of the coatings preparation process.

**Figure 3 materials-13-00629-f003:**
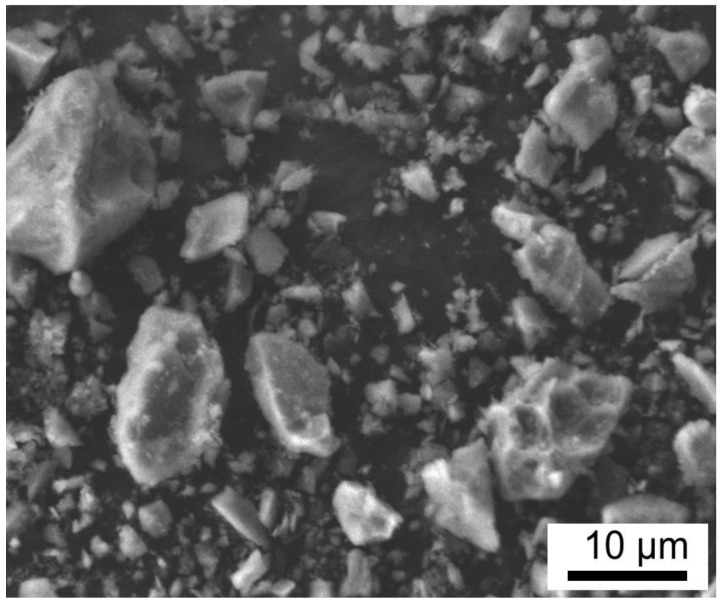
SEM of AYZ powder.

**Figure 4 materials-13-00629-f004:**
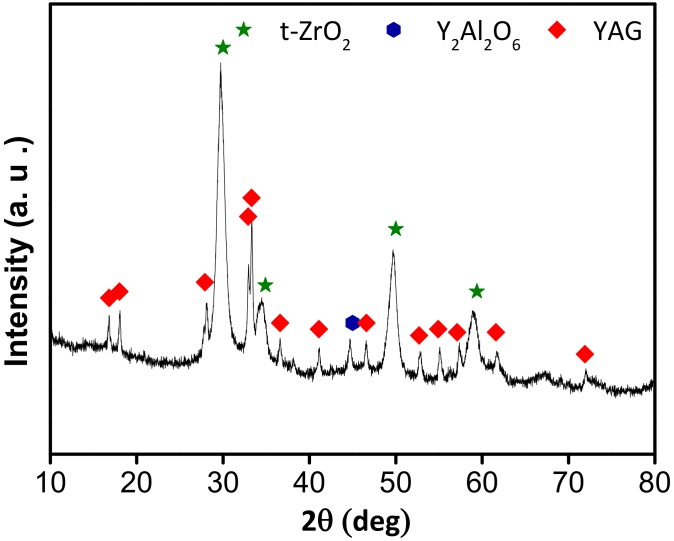
The X-ray powder diffraction (XRD) pattern of AYZ powder after calcination at 850 °C.

**Figure 5 materials-13-00629-f005:**
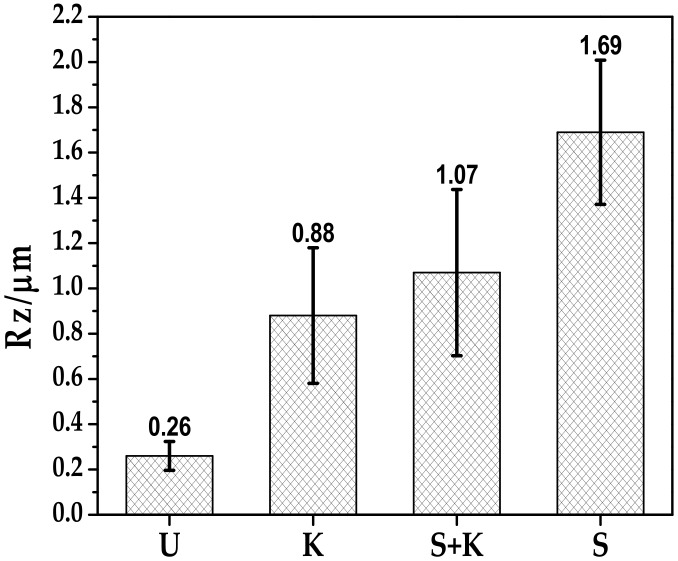
The roughness after pretreatment of stainless steel (U – ultrasonic cleaning, K – chemical etching with Kroll’s reagent, S – sandblasting, S+K – sandblasting + chemical etching with Kroll’s reagent).

**Figure 6 materials-13-00629-f006:**
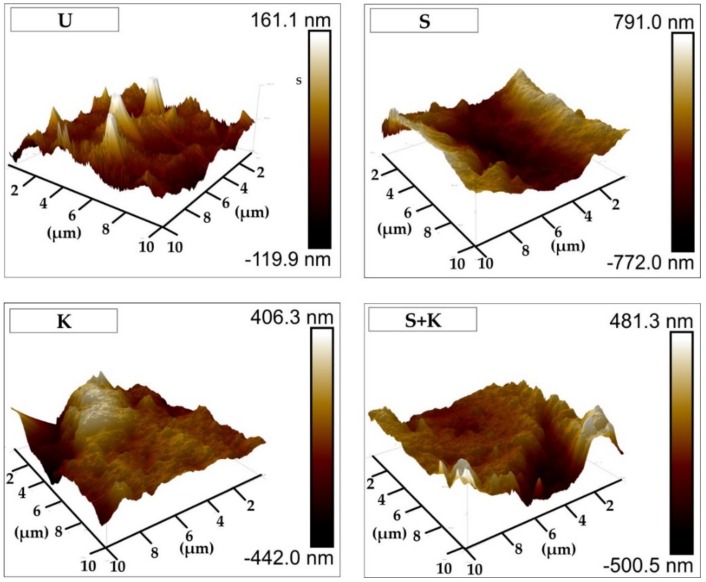
Atomic force microscopy (AFM) images representing roughness of the stainless steel surfaces treated by different cleaning procedures. The applied cleaning procedure is indicated by the abbreviation in the upper left corner of each figure (U – ultrasonic cleaning, K – chemical etching with Kroll’s reagent, S – sandblasting, S+K – sandblasting + chemical etching with Kroll’s reagent).

**Figure 7 materials-13-00629-f007:**
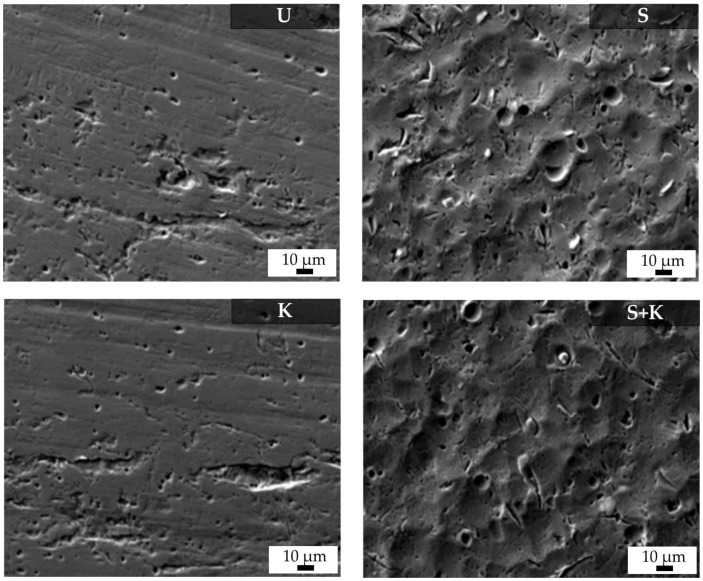
SEM micrographs of the stainless steel surfaces treated by different cleaning procedures. The applied cleaning procedure is indicated by the abbreviation in the upper right corner of each figure (U – ultrasonic cleaning, K – chemical etching with Kroll’s reagent, S – sandblasting, S+K – sandblasting + chemical etching with Kroll’s reagent).

**Figure 8 materials-13-00629-f008:**
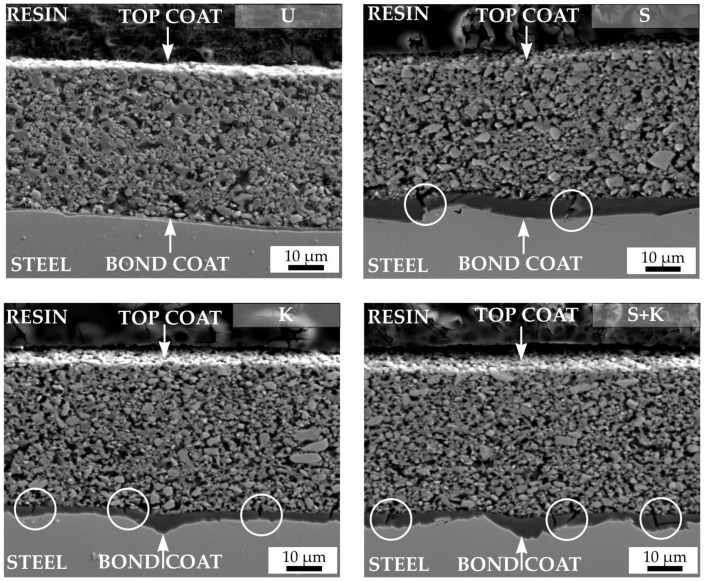
SEM cross sections of pyrolysed C2c coatings. The applied cleaning procedure is indicated by the abbreviation in the upper right corner of each figure (U – ultrasonic cleaning, K – chemical etching with Kroll’s reagent, S – sandblasting, S+K – sandblasting + chemical etching with Kroll’s reagent).

**Figure 9 materials-13-00629-f009:**
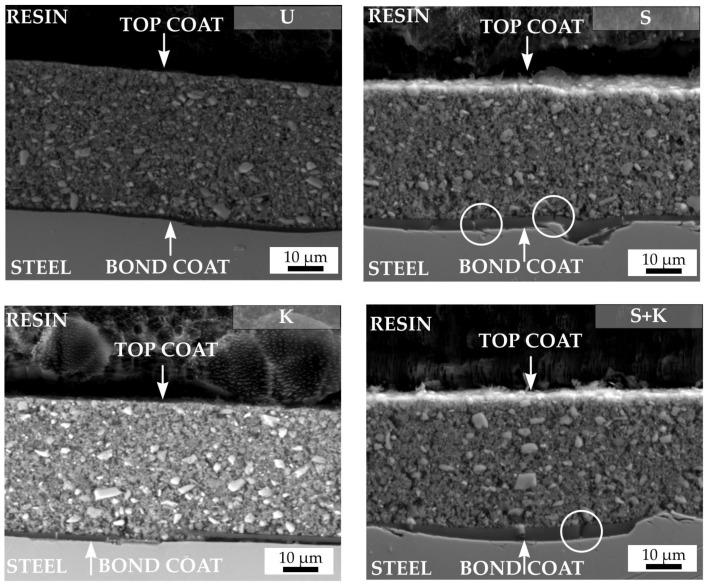
SEM cross sections of pyrolysed D2-PP coatings. The applied cleaning procedure is indicated by the abbreviation in the upper right corner of each figure (U – ultrasonic cleaning, K – chemical etching with Kroll’s reagent, S – sandblasting, S+K – sandblasting + chemical etching with Kroll’s reagent).

**Figure 10 materials-13-00629-f010:**
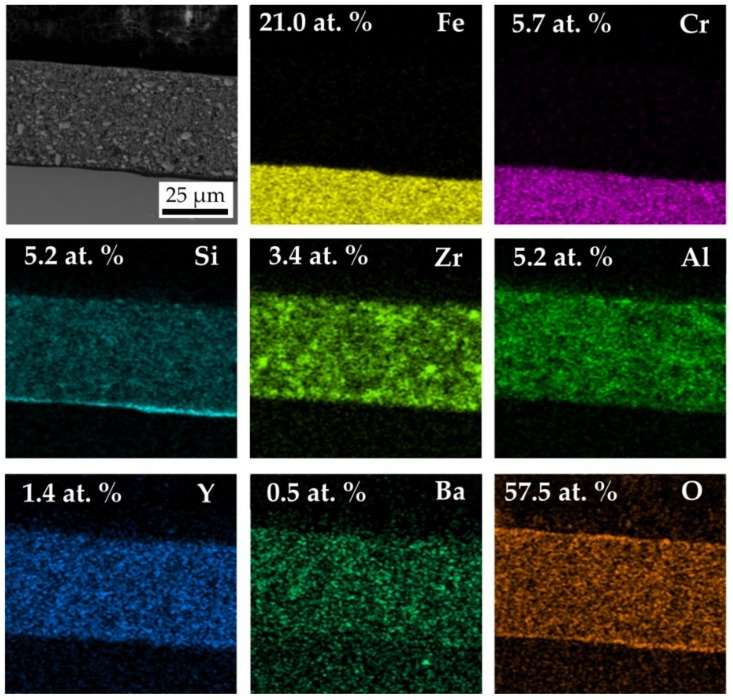
SEM/energy-dispersive X-ray spectroscopy (EDX) of D2-PP coatings documenting distribution of elements in the coating and in substrate.

**Figure 11 materials-13-00629-f011:**
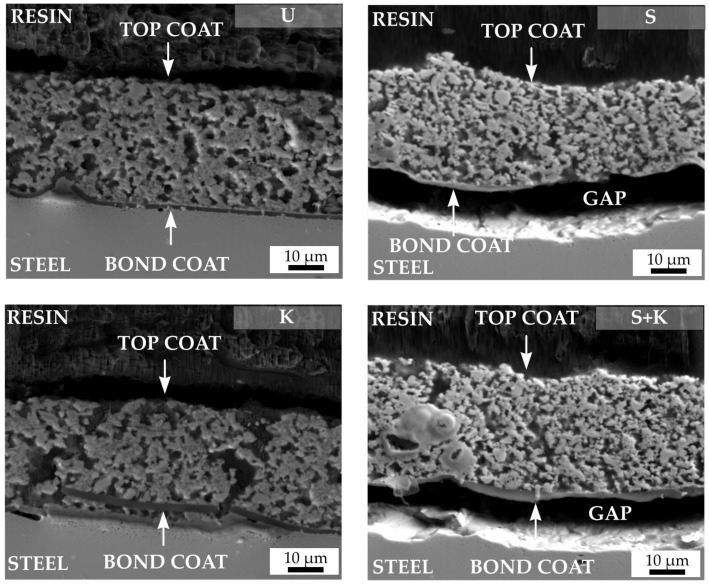
Cross sections of C2c coatings after oxidation tests. The applied cleaning procedure is indicated by the abbreviation in the upper right corner of each figure (U – ultrasonic cleaning, K – chemical etching with Kroll’s reagent, S – sandblasting, S+K – sandblasting + chemical etching with Kroll’s reagent).

**Figure 12 materials-13-00629-f012:**
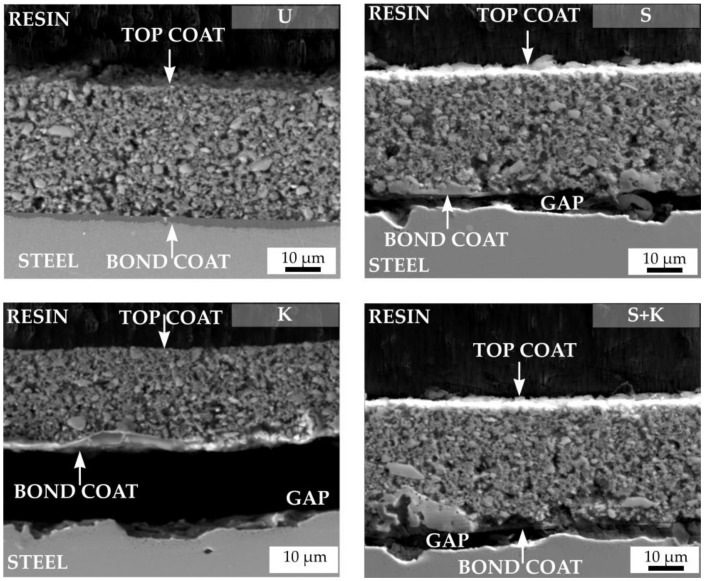
Cross sections of D2-PP coatings after oxidation tests. The applied cleaning procedure is indicated by the abbreviation in the upper right corner of each figure (U – ultrasonic cleaning, K – chemical etching with Kroll’s reagent, S – sandblasting, S+K – sandblasting + chemical etching with Kroll’s reagent).

**Table 1 materials-13-00629-t001:** The description of cleaning procedures.

Symbol	Pretreatment of Stainless Steel
U	3-step ultrasonic vibration cleaning in acetone, ethanol and deionised water (10 min each)
S	Sandblasting with glass beads (70–110 μm), ultrasonic cleaning in deionised water
K	Chemical etching—Kroll’s reagent (deionised water, HNO_3_, HF), 20 s
S+K	Sandblasting with glass beads + etching with Kroll’s reagent

**Table 2 materials-13-00629-t002:** Basic properties of filler materials.

Passive Fillers	d50 (µm)	*ρ* (g/cm^3^)	CTE (10^−6^/K)
8YSZ	0.5	6.1	11.5
AYZ	1–10	4.6	8.6
G018-281	0.5–5	2.7	12.1

**Table 3 materials-13-00629-t003:** Compositions of the composite top coats after pyrolysis (vol. %).

Sample Name	Durazane 1800	8YSZ	AYZ	G018-281	CTE (10^−6^/K)
C2c	30	35	-	35	10.1
D2-PP	25	20	20	35	9.6

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
