# Peer review of "PDC Glass/Ceramic Coatings Applied to Differently Pretreated AISI441 Stainless Steel Substrates"

_materials, 2020, doi:10.3390/ma13030629_

Round 1
Reviewer 1 Report
After carefully reviewing the “PDC glass/ceramic coatings applied to differently pre-treated AISI441 stainless steel substrates”. As the abstract mentioning, the U-type pre-treatment is dominated the bonding strength between coating layer and AISI441 substrate. There are two polymeric derived ceramic (PDC). Two layer-coating method is employing with interface of Durazane 2250 bond coat and with Durazane 1800 added with passive filers and commercial glass later. The characterization methods are SEM/EDX for cross-sectional morphologies and elements mapping in the region of coatings and substrates , AFM for coating surface roughness after cleaning methods applied on the AISI441 substrates. Furthermore, authors show the static thermal corrosion test for the aforementioned specimens.
However, there are some points should be addressed as following:
Obviously ultrasonic method is lowest energy input on the substrate in the surface treatments as authors defined as clean methods. What the main purpose is authors want to emphasize in clean method? And the surface roughness in Figure 5 should be added the error bar. What is the method or equipment for measuring CTE? Please give the brane/type information. What is the spray method? Nozzle geometry? Air-pressure? What is the elemental composition in bond coat? And the bonding strength for Durazane 2250 and two-layer coatings? The line 246-247, “ Fe and Cr, as the …., were detected below the protective bond coat.” The Cr depleted from AISI441 substrate in the surface is spontaneously forming the protective oxide layer. As my view, the Cr depleted in the bond coat and the Zr, Si, Y and Ba penetrated in the bond coat. And please give atomic percentages for aforementioned elements. Please check Figure 9 carefully.
Reviewer 2 Report
The authors investigate an interesting topic on the influence of the pre-treatment of a stainless steel surface pre-treatment for further application of a polymer-derived ceramic coating. This study has important practical implications. It is interesting that the U-treatment (I suppose, untreated) gives the best results in terms of the coating bonding. Usually, people would prefer sandblasting or etching.
Thу paper can be published after the following questions are addressed.
For the XRD patterns, what is the crystal type for ZrO2? m, t, с ? In Fig. 5 the tolerance intervals are needed. Also, it is unclear what roughness parameter was used (Rz, Rz, etc., please indicate). Line 192 - Average roughness is Ra, not Rz, please clarify. Is it the steel surface roughness that generates the coating delamination? Have you happened to grit the surface to Rz (or Ra???) 1.69 instead of sandblasting and test the coating for the sake of comparison of the methods, not the surface roughness? Or maybe it is the thin nanoscale oxide layer composition on stainless steel that contributes towards the coating adhesion? Please make some comments on that in the discussion. The Materials and Methods section must be revised so that a Corrosion test section appears separately from the Characterization methods. P. 2.4 mentions Raman spectroscopy. Please either remove the method or include the Raman spectra and their discussion; this could probably address my point 4. AFM figures can nicely complement Fig. 6 and show the roughness there. Otherwise, please do not mention the methods whose results you do not show in the paper.Author Response
Please see the attachment.

Round 2
Reviewer 2 Report
Fig. 6. x,y scale is missing; it is mandatory to have it. If possible, z scale should be the same for all the images.
